# Effect of Intertidal Vegetation (*Suaeda salsa*) Restoration on Microbial Diversity in the Offshore Areas of the Yellow River Delta

**DOI:** 10.3390/plants13020213

**Published:** 2024-01-11

**Authors:** Zhaohua Wang, Kai Liu

**Affiliations:** 1First Institute of Oceanography, MNR, Qingdao 266061, China; wangzhaohua@fio.org.cn; 2Dongying Research Institute for Oceanography Development, Dongying 257000, China

**Keywords:** vegetation restoration, *Suaeda salsa* plantation, high-throughput detection, bacterial diversity, archaeal diversity

## Abstract

The coastal wetlands in the Yellow River Delta play a vital role in the ecological function of the area. However, the impact of primary restoration on microbial communities is not yet fully understood. Hence, this study aimed to analyze the bacterial and archaeal communities in the soil. The results indicated that *Marinobacter* and *Halomonas* were predominant in the bacterial community during spring and winter. On the other hand, *Muribaculaceae* and *Helicobacter* were prevalent during the core remediation of soil, while *Inhella* and *Halanaerobium* were predominant in non-vegetation-covered high-salinity soil. The bacterial Shannon index showed significant differences in vegetation-covered areas. For archaea, *Salinigranum*, *Halorubrum*, and *Halogranum* were dominant in vegetation areas, while *Halolamina*, *Halogranum*, and *Halorubrum* were prevalent in non-vegetation areas. The colonization of *Suaeda salsa* led to differences in the composition of bacteria (22.6%) and archaea (29.5%), and salt was one of the significant reasons for this difference. The microflora was more diverse, and the elements circulated after vegetation grounding, while the microbial composition in non-vegetation areas was similar, but there was potential competition. Therefore, vegetation restoration can effectively restore soil ecological function, while the microorganisms in the soil before restoration provide germplasm resources for pollutant degradation and antimicrobial development.

## 1. Introduction

Intertidal vegetation was the primary producer in the coastal ecosystem [1]. Retaining coastal soil water, minimizing erosion, and enriching faunal types are important. However, the dramatic expansion of mariculture and fishery production activities has severely degraded intertidal plant habitats. Degraded vegetation exacerbates intertidal surface exposure, and sunlight exposure will exacerbate the salinization of intertidal soils, destroying the ecological function of soils and damaging microbial environments [2,3]. It could ultimately transform the coastal soil environment into salting beaches with low plant diversity [4,5]. Furthermore, the degradation of vegetation also leads to a decrease in the ability of the intertidal zone to absorb pollutants and heavy metals in the sedimentary environment, which exacerbates environmental pollution in the intertidal zone. Therefore, restoring intertidal vegetation is an important biological approach to protect the intertidal ecosystem [6].

*Suaeda salsa* is the major component of intertidal plant habitats in Northeast Asia [7]. It could grow with the help of internal water by increasing self-tolerance to the salinity in nearshore tidal flats. The shadows created by saltmarsh plants could prevent direct sunlight on the coast, reducing transpiration and helping desalinate the soil from precipitation, tides, and river flushes. Moreover, *Suaeda salsa* could also significantly increase the soil’s microbial diversity and improve metabolic function by improving the processes of the habitat [7,8]. Even after being planted for one year, *Suaeda salsa* still benefited coastal land by reducing saltness, preventing degradation, and increasing bacterial diversity [7]. In particular, at different stages of vegetation colonization, the soil’s total phospholipid fatty acid content increased, which affected the total content and composition of microorganisms. In addition, the death of vegetation could promote an increase in organic matter in the soil. The decomposition could enrich nutrients and bacterial and archaeal communities, and the increased flora could promote the spread of saline *Suaeda salsa* colonization [9]. These behaviors can form a positive cycle and improve the ecological function of coastal wetlands.

However, the nearshore tidal flats where only *Suaeda salsa* grows with a poor environment, are high susceptibility to human activities [10]. This behavior prolongs the exposure of surface soil to direct sunlight and ultimately results in a *Suaeda salsa* coastal area. Moreover, it reverses the positive process from the rooting of *Suaeda salsa* to vegetation covering. For this reason, the Chinese government has carried out large-scale artificial plant habitat restoration in the northern intertidal zone. However, the self-sustainability and evaluation criteria for vegetation restoration are still unknown. Microorganisms are the basis for undertaking biogeochemical elemental cycling [11]. The change in microorganism communities after restoration could serve as an important criterion for measuring restoration. Intertidal plant restoration in the Yellow River Delta is one of the most successful cases of intertidal vegetation restoration in northern China. After successive restorations from 2018 to 2021, intertidal vegetation in the restored areas of the Yellow River Delta has been restored to a certain extent. Therefore, this study analyzes the plant replanting status of restored areas via the normalized vegetation index (NDVI). Furthermore, we explore the different effects of vegetation rooting conditions with respect to year-round changes in the community structure of surface soil microorganisms. Our study provides data and theoretical support for guiding vegetation restoration and improving the success of plant restoration in coastal areas.

## 2. Results

### 2.1. Normalized Vegetation Index

In order to monitor the changes in vegetation restoration in the restoration area, we selected the month of August as the monitoring period. This is because the most luxuriant growth of *Suaeda salsa* takes place during August. The normalized vegetation index (NDVI) for this study area was analyzed (Figure 1). The changes in NDVI from 2019 to 2023 are shown in Figure 1. In 2019, the NDVI of the restoration area was 0.14 to 0.18, and the areas with the most luxuriant growth were located at stations S3 and S5. The lowest NDVI was observed at the supratidal station (S1) and the station closer to seawater (S6), with vegetation indexes of 0.14 and 0.15, respectively. In 2020, the vegetation in the restored area did not change significantly due to a disturbance from ecological restoration activities. In 2021 and 2022, as the ecologically restored vegetation entered the self-sustaining stage, the NDVI showed a significant increase, with an average growth rate of 0.03/a. In 2023, the vegetation index of the restored area was increased to 0.17~0.26, and the stations with more luxuriant growth changed to S1–S4. The average NDVI increased to 0.22 ± 0.03, and the lush vegetation area expanded. The area with the least vegetative growth changed from S1 to S6. It was also observed that the artificial vegetation ecological restoration project in the coastal zone played a certain role in the restoration of coastal vegetation habitats. From 2019 to 2023, the total vegetation index of the restoration area increased from 0.96 to 1.34, with an annual rate of increase of 0.08/a. However, the RSD change relative to changes in the index of the different zones reduced from 8% to 5% and then increased to 13%, indicating a clear differentiation in vegetation restoration after self-sustainment.

Relative to the restoration areas, NDVI recovery in the S1 and S4 areas increased most significantly from 0.14 to 0.24 and from 0.16 to 0.26, respectively. Except for stations close to seawater, the difference in vegetation index between stations gradually decreased. This may be caused by the ecological restoration of vegetation, leading to the expansion of *Suaeda salsa* seeds from the coastal zone to the supra-tidal area with rising and falling tides. In contrast, the NDVI recovery of S5 and S6 was slow due to their proximity to the estuarine region, and these areas were susceptible to rising tides. From the recovery rate, the vegetation recovered the fastest from 2020 to 2021, and the recovery rate slowed down in 2022 and 2023. However, during these two years, the average NDVI did not change, and the distribution of vegetation exhibited even and lush settings, which increased significantly from that before the restoration. This indicated that *Suaeda salsa* habitats in the restored area of the Yellow River Delta had entered a more stable self-sustaining stage after ecological restoration. Therefore, different spatially existing vegetation coverages can be used to explain the microbial diversity changes in the topsoil before and after temporal remediation.

### 2.2. pH and Salinity

The pH and salinity in soil samples in the four seasons are shown in Table 1. In all four seasons, the pH of most soil samples exceeded 8.1. Soil pH was highest during the spring and autumn seasons, with the lowest values occurring at the station that was most onshore and had an intermittent freshwater input (S1): 7.87 to 7.96. In the areas subject to seawater inundation year-round (S4–S6), the pH ranged from 8.27 to 8.32. In the shoreline areas with a lower frequency of inundation, such as S1 and S2, the pH ranged from 7.92 to 8.36. There was no significant relationship between pH and vegetation cover, but the lowest pH was found in the shoreline areas subject to intermittent freshwater: ~7.87. This suggested that freshwater inputs are an important factor influencing intertidal pH changes in the region. Salinity varied between 30.62 and 57.32, with the highest salinity reaching ~56 (S1) in the exposed area. The lowest salinity was found in the stations subject to regular seawater inundation and covered by *Suaeda salsa* (S5 and S6), with mean values of 30.20 and 30.90, respectively. Salinity was also lower in areas subject to riverine influences: ~32.89. This phenomenon suggests that seawater and riverine scouring are also important factors influencing the salinity of the soil in both supratidal and intertidal zones. However, in other areas, the salinity of stations covered by *Suaeda salsa* vegetation (30.18~44.63) was significantly lower than that of the unvegetated area, even though the density of *Suaeda salsa* in this area was low, which suggests that *Suaeda salsa* in the coastal zone plays an important role in slowing down the salinization of intertidal soils.

### 2.3. Bacterial Diversity

High-throughput bacterial assays revealed (Figure 2) that the spring restoration area was dominated by *Marinobacter* (1.08–20.27%), *Halomonas* (0.72–10.88%), *Inhella* (0.63–8.60%), and *Muribaculaceae* (1.14–10.21%). In the summer, it was dominated by *Psychrobacter* (27.21–65.74%), *Gillisia* (0.21–33.33%), *Gramella* (1.04–17.98%), and *Planococcus* (0.14–40.82%). In the autumn, it was dominated by *Psychrobacter* (3.88–73.51%), Planococcus (0.34–4.93%), *Gillisia* (0.66–4.46%), and Gramella (0.27–8.87%). In the winter, it was dominated by *Marinobacter* (0.07–16.66%), *Methylophaga* (0.22–5.66%), *Woeseia* (0.30–3.99%), *Roseovarius* (0.67–4.95%), and *Loktanella* (0.45–4.16%). However, there were still differences between sampling sites.

Regarding post-vegetation rooting, the highest number of genera with relative abundance exceeding 10.00% was found in post-vegetation rooted topsoil (summer), followed by autumn, spring, and winter. For bacteria, *Psychrobacter* exceeded 10.00% in the topsoil in both summer and autumn seasons and exceeded 70.00% at a few sample stations. The results of unrooted surface soil were similar to those of rooted soil. The number of dominant genera relative to each season with a relative abundance of more than 10.00% was as follows: summer, autumn, spring, and winter. *Marinobacter* was dominant in spring and winter at different sample sites, while *Psychrobacter* had some dominance in summer, autumn, and winter. The above results indicate that *Psychrobacter* is common in coastal wetland environments and is strongly influenced by temperature. *Marinobacter* is more likely to grow in topsoil without rooting environments, which is presumably related to continuous seawater inundation in the coastal restoration area. Furthermore, the rooted vegetation partially alleviated the competition for bacterial populations and contributed to the diverse bacterial community’s composition.

### 2.4. Archaeal Diversity

The high-throughput archaeal results are shown in Figure 3. Spring archaea were dominated by *Salinigranum* (6.53–34.16%), *Halorubrum* (12.07–29.32%), *Halogranum* (1.43–26.86%), *Natronomonas* (2.82–6.40%), and *Halolamina* (1.75–20.24%); summer archaea were dominated by *Halorubrum* (1.35–13.34%), *Nitrososphaeraceae* (1.74–37.59%), *Bathyarchaeia* (1.01–21.89%), *Halolamina* (1.38–9.73%), and *Haloferax* (1.39–11.28%). *Halorubrum* and *Halolamina* were also common dominant archaea in the spring, and the relative abundance of other dominant archaea in the spring decreased to varying degrees during the summer. In the autumn, the dominant archaea were *Nitrososphaeraceae* (0.72–21.68%), *Natronomonas* (2.57–9.12%), *Salinigranum* (3.68–16.38%), *Bathyarchaeia* (0.46–20.15%), and *Halorubrum* (2.21–21.21%). *Nitrososphaeraceae, Bathyarchaeia*, and *Halorubrum* were also common dominant genera in the autumn. In the winter, all soil samples were mainly dominated by *Nitrososphaeraceae* (0.91–27.75%), *Halorubrum* (1.16–12.51%), *Bathyarchaeia* (0.51–15.37%), *Salinigranum* (1.19–15.66%), *Halogranum* (2.29–5.52%), and *Haloferax* (0.78–7.20%). *Nitrososphaeraceae, Halorubrum, Bathyarchaeia*, and *Salinigranum* were also among the common autumn archaea, and the relative abundance of the other autumn dominant genera did not change significantly.

Relative to plant rooting, the archaeal side exhibited the highest number of species, with relative abundances greater than 10.00% in the spring, followed by the winter, summer, and autumn seasons. The number of dominant archaea with relative abundances greater than 10.00% was also higher in the spring and winter seasons than in the summer and autumn seasons in the unrooted area. These results indicate that the relative abundance of archaea differed significantly during different seasons, but some of the dominant archaeal genera did not change. For example, *Salinigranum*, *Halorubrum*, and *Nitrososphaeraceae* were dominant in different seasons and different regions, but the relative abundance did not exceed 10.00% in all samples.

### 2.5. α Diversity of Bacteria

The box plots showed that the number of observable species (observed species), Chao index (Chao1), and Shannon index (Shannon) varied during different seasons, but the trends were close to each other (Figure 4). This showed that there were no significant differences in observed species and the Chao1 and Shannon indexes between groups (*p* > 0.05), but there were differences within groups. The largest difference in the number and abundance of observable bacteria was found in RU without vegetation rooting, and the smallest difference was found in HSU. The smallest difference in observed species and the Chao1 index was found in RR in the area with rooting. This shows that vegetation rooting effectively improved the bacterial habitat in the topsoil layer, and the bacterial population was enriched relative to a specific direction. Moreover, the vegetation rooted at the riverside mitigated the changes in surface soil introduced by water flow. This resulted in insignificant differences with respect to bacterial number and abundance between RU and RR; thus, the diversity of bacterial groups in RR was higher but the degree of within-group variations was smaller. Interestingly, HSU and HSR in high-salt environments presented differences with respect to bacterial diversity, with HSU being more stable and concentrated compared to HSR. This result suggests that the high salt content was the environmental factor that directionally enriched some microorganisms. However, vegetation effectively mitigates this effect, increasing bacterial abundance and diversity in the surface soil.

The observed species and Chao1 were higher in samples from non-rooted and winter-rooted areas during the spring season, followed by samples from rooted areas during spring and non-rooted areas during winter. Meanwhile, the observed species and Chao1 intra-group differences were higher for the spring, autumn, and winter seasons. In the non-rooted zone, intra-group differences were smaller for the summer non-rooted zone, and the rooted zone did not have much variation in the intra-group differences across seasons. This result suggests that the high summer temperature, precipitation, and high salinity suppress the number and abundance of flora. Vegetation rooting effectively mitigated the differences in surface soil bacterial numbers and abundance caused by climate, but the effect was not significant (*p* > 0.05). Relative to bacterial diversity, the autumn-rooted sample differed significantly (*p* < 0.05) from the spring non-rooted and the winter-rooted samples. This suggests that vegetation growth in summer provided a prolonged and effective promotion of topsoil alteration that triggered shifts before and after rooting.

### 2.6. α Diversity of Archaea

The α diversity of the archaea revealed that the observed species, Chao1, and Shannon of the archaeal community of spring samples were higher than the other seasons (Figure 5A). Meanwhile, the species diversity (Shannon index) was higher in the summer and autumn samples. The differences in archaea from rooted and non-rooted samples and relative to different seasons were evaluated using archaeal α diversity analysis (Figure 5E,F). The results revealed that spring-rooted planting samples exhibited higher values than the other seasons in terms of the number of observable species, species richness, and Shannon index. However, only the number and diversity of archaea were significantly higher than that of the summer samples (*p* < 0.05). Meanwhile, in non-rooted soil, spring samples were also higher than other samples, but the difference was not significant. Combining all the results, the soil vegetation after rooting was able to provide organic matter for some of the archaea. It could target the enrichment of some archaea, which resulted in differences in the number and diversity between spring and summer. These results suggest that the soil flora of intertidal surface soil is enriched rather than diversified after restoration. However, this process requires longer and continuous inputs and is more affected by the season. Moreover, bacteria were more susceptible than archaea to environmentally induced changes, including higher within-group dispersion. This phenomenon suggests that bacteria assume more environmental functions after vegetative rooting and that archaea show more tolerance than adaptation after vegetative rooting.

### 2.7. Principal Coordinate Analysis for Bacteria and Archaea

Via PCoA analysis (Figure 6), the total explained amount of differences in bacterial communities between samples was 47.9% on the first and second principal axes. The differences between spring samples and other samples, between winter-rooted areas and summer samples, and between different autumn-rooted areas were mainly reflected on the first principal axis with an explanation of 35.8%. The differences between winter-rooted areas and different samples of non-statistical seasons and the differences between spring and other season samples were reflected in the second main coordinate axis with an explanation of 12.1%. Meanwhile, there were three relatively separated zones in the vertical direction, i.e., the second coordinate axis, throughout the PCoA analysis—namely, the winter-rooting zone samples; the other samples during summer, autumn, and winter; and the spring samples. This result suggests that there is a large difference between the bacteria in spring surface soil and the other seasonal samples and that rooting or not rooting has a weaker effect on soil bacteria in spring. Rooting in winter had a greater effect in terms of surface soil bacterial communities, and the seasonal effect was not as pronounced as the effect of rooting in terms of surface soil. Therefore, surface soil bacteria were affected by both seasonal and rooting conditions, but more so by seasonal effects.

Via PCoA analysis (Figure 7), the total explained amount of archaeal community differences between samples in the first and second principal axes was 36.4%. The differences between rooted and non-rooted samples during the summer and autumn seasons were mainly reflected in the first principal axis, with an explained amount of 23.1%. The differences between rooted and non-rooted samples in the spring were mainly reflected in the second coordinate axis, with an explained amount of 13.3%. Meanwhile, there were three relatively separated areas in the PCoA analysis: spring samples and rooted and non-rooted zone samples in summer, autumn, and winter. The latter two did not unfold thoroughly on the first and second coordinate axes. This result suggests significant archaeal community differences in spring surface soil compared to soil from the other seasons.

### 2.8. Venn Analysis of Seasonal and Rooting Conditions

Venn analysis based on seasons and rooting conditions revealed that the number of bacterially shared OTUs among samples was only 139 (Figure 8). The highest number of bacterial OTUs in samples from spring non-rooted areas was 12,470, and the lowest was observed in rooted areas during the autumn at 736. The difference between rooted and non-rooted areas was insignificant, except for non-rooted areas in the summer, where the OTUs were 43.89%. This indicates that the season is an important factor affecting the number of OTUs in the surface soil. However, vegetation rooting can significantly interfere with the effect of this factor in terms of the number of OTUs in the surface layer of bacteria. Meanwhile, bacterial OTUs were significantly reduced in the surface soil during summers after rooting. This may be because vegetation rooting provided better habitats for some of these bacteria when under environmental influences, such as high temperatures and precipitation. Furthermore, a more pronounced directional enrichment of some surface bacteria also occurred.

The number of archaeal OTUs among samples was only 147, with the highest number of archaeal OTUs being 9628 in spring non-rooted samples (Figure 9). The lowest number of OTUs was observed in autumn-rooted areas at about 1845. The number of archaeal OTUs was lower in spring- and autumn-rooted areas compared to that of the non-rooted area. However, for summer- and winter-rooted samples, the number of OTUs was higher in non-rooted areas. This suggests that seasonal changes and the root-planting restoration of *Suaeda salsa* in coastal wetlands are the main factors affecting surface soil archaea. The archaea are more tolerant but less metabolically efficient. After entering the summer seasons, the vegetation provided more organic matter and a suitable living environment for the surface soil, which provided the possibility of the diversification of archaea. However, the bacteria were more diverse compared to the archaea; thus, the bacteria appeared to have a directional enrichment effect within the same habitat. After entering the fall season, the temperature dropped and the vegetation exhibited weakened growth, which resulted in a reduction in organic matter available for the archaea in the rooted area. Thus, the number of dominant archaea in the vegetated area was reduced. Although the reduction in temperature and precipitation acted as the main driver that screened out some archaea, vegetation still provided a stable environment for the archaea. This resulted in a higher number of archaeal OTUs in the vegetated zone.

## 3. Discussion

### 3.1. The Process of Suaeda salsa Restoration

Relative to the annual change in satellite vegetation indexes, after the artificial restoration in 2019–2021, the overall recovery of vegetation in the restoration area was observed in 2023 (Figure 1). The average vegetation index of the restoration area increased by 37.58%, and the total vegetation index increased by nearly 40%. This indicates that the artificial ecological restoration project has obvious effects on the reconstruction of vegetation habitats in the coastal zone [12,13]. However, on the whole, the pattern of vegetation restoration in different areas and time periods is obvious. During the first 2 years of the restoration process, the influence of human restoration activities, such as land preparation, plant transplantation, and seeding, was greater. The restoration process of vegetation in the restored area was slow, and some stations even appeared to be on the decline (S3). Vegetation restoration was mainly carried out at stations S2–S5. After 2 years, the restoration area accumulated a large amount of *Suaeda salsa* seeds, and after tidal activities, *Suaeda salsa* seeds spread to all stations within the study area. As a result, the NDVI of all stations increased significantly. In 2022, after the seed dispersal process in the previous three years, the entire study area had a peak *Suaeda salsa* restoration period, and the NDVI of all stations increased significantly. When entering the 2nd year of self-sustaining vegetation (2023), NDVI indices at shoreline stations (S1) showed a significant increase, but few changes were observed in the core restoration area (S6). This station (S6) was easily inundated by seawater, suggesting that natural conditions such as tides continue to have a significant impact on the habitat of *Suaeda salsa* plants [14,15].

According to changes in the vegetation index, ecological restoration processes relative to promoting *Suaeda salsa* habitats in coastal areas are shown in Figure 3. Firstly, via manual restoration, *Suaeda salsa* plants or seeds were planted or sown in the restored area to increase the number of seeds. Then, after a period of manual intervention, the restored area gradually recovered. Rehabilitated vegetation continues to produce many seeds each year, and they are dispersed in the core restoration area [16,17]. With tidal activity, precipitation, and animal transport, the seeds gradually expand to other areas, restoring the entire area and surrounding areas [18,19]. However, seawater-induced declines in rehabilitated vegetation also suggest that restored areas must be replenished with seed sources. Long axial timescales for complementary technological processes are necessary in order to ensure that vegetation habitats in coastal zones continue to recover until they become self-sustaining.

### 3.2. Characterization of Bacterial Changes in Soils from Vegetation Restoration Areas

As for bacteria, *Marinobacter,* an aerobic Gram-negative bacterium, was more common in spring and winter seasons, and its relative abundance was higher in high-salt or emergent-only topsoil [20]; this is consistent with its general salinity adaptation: i.e., the salinity required for the growth environment was 0.08–3.5 M. The results showed that *Marinobacter* was more tolerant to high temperatures but less tolerant to low temperatures in spring and winter seasons. At the same time, the genus had good high-temperature adaptation but poor low-temperature tolerance, which was not consistent with the results of the higher relative abundance in spring and winter seasons; however, this was consistent with the lower relative abundance in summer and autumn seasons, suggesting that abundant animal activities in summer and autumn in conjunction with warmer temperatures brought rich organic substrates to the surface soil, providing more possibilities for other microorganisms [20]. At the same time, precipitation during the same period and heat diluted the salinity of the topsoil, which was unfavorable to the growth of *Marinobacter*. In contrast, the lack of precipitation in spring and winter and intertidal recharge introduced salinity and some heat exchange to the surface soil in order to assist *Marinobacter* growth, while less animal activity reduced organic matter contents and diversity in the surface soil of the restoration area, resulting in an increase in the relative abundance of *Marinobacter*. Similarly, to *Marinobacter,* the relative abundance of *Halomonas* was higher in the spring, especially in high-salt environments, unlike the relative abundance of *Halomonas*, which was lower in the winter [21,22]. As Gram-negative aerobic microorganisms, they exhibit higher salt acclimatization: 0.1–32.5% (*w*/*v*). It is inferred that the elevated salinity is more favorable for the growth of salt-tolerant microorganisms compared to the winter and spring seasons when there is less precipitation and increased seawater influences on the surface soil; however, low contents in the winter and slow growth in the spring due to the lack of organic matter resulted in a lower relative abundance compared to *Marinobacter*.

At the same time, *Halomonas*, as a bacterium with flagella and other characteristics, has a certain ability to move, resulting in its more abundant content in seawater; thus, there is a possibility of rapid inoculation and propagation in surface soil in the spring season, resulting in a rapid increase in relative abundance during the transition from winter to spring. *Inhella* and *Muribaculaceae* are also microorganisms with higher relative abundances in the surface soil in the spring [23,24]. *Inhella* is commonly found in freshwater environments with significant eutrophication, while *Muribaculaceae* is a common gut microorganism, which suggests that the increase in human activities during spring led to an increase in associated metabolizing microorganisms, but this was limited by the high salinity of the environment, which was not favorable for an increase in non-tolerant genera and thus resulted in a relative abundance that was not as high as that of *Marinobacter* and *Halomonas*; this also largely ceased to occur in subsequent seasons. *Psychrobacter*, *Gillisia*, *Gramella*, and *Planococcus* had higher relative abundances in surface soils in the summer and autumn seasons, and the associated genera were all either cold-tolerant or extremely cold-tolerant, with *Psychrobacter* being observed in a variety of organ types and possibly multiple species of animal organs [25,26,27,28]. *Psychrobacter* is found in many types of animal organs and may be an opportunistic causative agent of a wide range of diseases, and *Gramella* is a macromolecular organic-matter-degrading bacterium in the ocean. Despite the good cold tolerance of these genera, their optimal growth temperatures are still around 25 °C. Compared to *Marinobacter*, *Halomonas*, and *Inhella* genera, the energy acquisition of *Gramella* may be competitively limited, the need for carbon sources and cold hardiness may be important in limiting the increase in their relative abundance in the winter intertidal zone. However, human and animal activities in the summer and fall provide abundant carbon source types in the intertidal zone, allowing them to outcompete other environmentally tolerant genera via large-scale growth and reproduction with associated organic molecules.

### 3.3. Characterization of Changes in Archaea in Vegetation Restoration Area Soil

Regarding archaea, *Salinigranum*, *Natronomonas*, *Halorubrum*, *Halolamina*, and *Halogranum* were present in all seasons as genera with high relative abundance [29,30,31,32,33]. Despite the variation in relative abundance, the degree of variation in relative abundance was lower than in bacteria. At the same time, the general salt tolerance of the archaea did not result in significant differences between the rooted and non-rooted areas of vegetation, and the differences between the areas could more likely come from differences in bacterial diversity, i.e., the small molecules produced by the bacteria after metabolizing the macromolecules, which provided metabolic substrates for the corresponding archaea but did not change the environment for archaeal survival. All the above genera are commonly found in high-salinity environments, such as the deep sea, salt lakes, and intertidal zones, and Dongying City generally experiences low precipitation in the winter and spring seasons, which leads to the salinization process of surface soil, including the evaporation of retained seawater after seawater tides and the evapotranspiration of seawater from the lower layer caused by a loss of surface water. In contrast, *Nitrososphaeraceae*, *Haloferax*, and *Bathyarchaeia* are uncommon in the spring environment and prevalent in the other three seasons, with *Bathyarchaeia* being prevalent in marine, freshwater, hydrothermal, hot spring sediments, and soils. However, there is no pure culture in the laboratory for the time being; thus, their physiological characteristics are not evident [34,35,36]. *Nitrososphaeraceae* is a chemoautotrophic archaeon that fixes carbon dioxide by oxidizing ammonia; thus, it is assumed that the reason for its low content in spring is that human and animal activities do not provide sufficient ammonia for the genus in the spring, which restricts the metabolism of the genus. The chemoenergetic heterotrophic bacterium *Haloferax* can produce antimicrobials, which can kill or organize the growth and reproduction of most other microorganisms, and it is an excellent germplasm resource for antimicrobial drugs. However, its relative abundance was not high compared to other dominant archaea, suggesting that antimicrobial production provides a specific competitive advantage for this archaeon in intertidal environments. However, it may still be insufficient compared to salt tolerance, broad substrates, and cold tolerance. Meanwhile, methanogens, as common archaea in the intertidal sediments of coastal wetlands with an extremely low relative abundance in this investigation, are extremely common in intertidal environments as the last link of carbon cycle degradation in anaerobic degradation environments [37,38]. This suggests that surface sediments may have been subjected to less direct inundation by seawater and rainfall, lacking an anaerobic environment. At the same time, since the bacterial community was not found to be rich in sulfate-reducing bacteria and they were analyzed in conjunction with low seawater inundation, the organic matter content in the surface soil was still at a relatively low level. This proves that salinity is the core influencing factor of the environment, and the restoration of *Suaeda salsa* planting has improved the surface soil, especially in summer. However, there is still room for improvement compared to salinization caused by the continuous transpiration of seawater. However, regarding vegetation restoration, the effect of anaerobic environments in the case of flooding is not considered for the time being, which favors the selection and application of salt-tolerant fungicides.

### 3.4. Phytoremediation Processes Altering Biodiversity in the Yellow River Delta

Unlike the composition of the bacterial community, which has two temporal compositions (winter–spring and summer–autumn), the archaeal composition has temporal differences between spring and summer–autumn–winter, but the differences are not obvious, and the related species are basically the same. Analyzing the effects of saltbush rooting in the bunker from a spatial perspective, we can find that planting saltbush has introduced restoration effects to the surface soil, but such restoration effects are not obvious in terms of salinization and temperature changes [15]. The main reason for this is that during the beginning of the restoration, the operation was not able to change the ecological environment and biological processes enough to accelerate the process of elemental cycling and energy transfer right now, which led to an increase in the diversity of bacteria in the summer season, further effectively degrading the richness of the organic matter in the soil and promoting the growth of some archaea; however, the persistence of the high environmental stresses of high salinity suppressed the more varied changes in the soil. In the *Suaeda salsa* planting area, the alkaline phosphatase activity on the surface soil layer was reduced, and the 10–20 cm layer increased the control, which was the predominant factor affecting soil bacterial community compositions [7]. Moreover, the aboveground parts of *Suaeda salsa* can extract salt, which dramatically decreases the topsoil layer’s salinity [39]. As a result, the planting of *Suaeda salsa* improved the soil properties and increased microbial communities. When summer heat and precipitation, growth cycles of *Suaeda salsa*, and the impact of animal migration cease, the organic matter content of surface soil decreases, leading to a decrease in bacterial diversity; simultaneously, the decrease in precipitation leads to an increase in the salinization of the surface soil, and high salt as an environmental pressure re-emerges as a central force in screening the surface soil, further reducing the diversity of the bacterial flora.

The restoration of intertidal salt flats has effectively alleviated soil salinization and seawater erosion and brought some organic matter to the surface soil directly and indirectly; moreover, it provided the possibility of soil desalination. However, the restoration’s duration was not sufficient, and the scale and growth of SALT are not optimistic, which does not essentially change the high-salinity environment in the intertidal zone; at the same time, bacterial diversity in the restored area is more affected by increased salinity and seasonal changes, which the salinization prevents the growth of *Suaeda salsa.* On the one hand, the influence of coastal dikes has led to the accumulation of seawater in the intertidal zone for a long period of time, which renders the intertidal zone salinized and unfavorable for the growth of salt flats; on the other hand, the scale and quality of flat cultivation both have room for improvement, and there is still the possibility of self-sustaining and positive cyclic succession in terms of bacterial and archaeal compositions, but it is still necessary to improve cultivation technology and increase the investment in cultivation to achieve self-sustainability.

## 4. Materials and Methods

### 4.1. Study Area

In this study, the Yongfeng Estuary restoration area was selected as the study area (Figure 10). According to the distribution of vegetation, the study area was divided into the following zones: supratidal no-vegetation zone (S1); supratidal vegetative zone (S2); intertidal vegetative zone (S3); intertidal no-vegetation zone in the Yongfeng River Estuary (S4); intertidal vegetative transition zone in the Yongfeng River Estuary (S5); the intertidal vegetative zone in the Yongfeng River Estuary (S6). Stations S3–S6 are the core areas of vegetation restoration, and the supratidal zone (S1–S2) is established via the shedding of seeds during plant restoration. Stations S1–S2 are near the tide embankment, which provides space for human activities. Therefore, these stations are more affected by human and animal activities. In addition, the rooted areas are shown as R/R, and ungrounded areas are shown as U/U. The four seasons are shown as C/C (spring, Chun), X/X (summer, Xia), Q/Q (autumn, Qiu), and D/D (winter, Dong).

### 4.2. Samples

In 2021, the artificial *Suaeda salsa* restoration project in the Yongfeng Estuary region entered the self-sustained recovery stage. Hence, this study used surface soil samples (1–5 cm) in the restored area that were obtained during spring, summer, autumn, and winter. A brief procedure was followed. Three replicates of vegetation and non-vegetation area samples were collected in the study during different seasons, from the supratidal no-vegetation zone (S1), the supratidal vegetative zone (S2), the intertidal vegetative zone (S3), the intertidal no-vegetation zone in the Yongfeng River Estuary (S4), the intertidal vegetative transition zone in the Yongfeng River Estuary (S5), and the intertidal vegetative zone in the Yongfeng River Estuary (S6). The corresponding sites were randomly selected according to the actual planting situation and satellite data. About 300 g of surface soil was collected using a sterile resin sampler. Plant limbs, stones, and other waste particles in the sample were removed. Sample soil was placed into sterile self-sealing bags for mixing. About 3 g of sediment samples was collected into 1.5 mL sterile centrifuge tubes (three parallel samples were collected simultaneously) and placed in a liquid nitrogen tank for temporary storage while the remaining samples were stored in a −20 °C incubator. The samples were stored in a −80 °C refrigerator until microbiological testing and the remaining samples were stored in a −40 °C refrigerator for physicochemical characterization.

### 4.3. Sample Analysis

pH and salinity: In this study, 10.0 g of a sample was transported into a 50 mL centrifuge tube; 35 mL of ultrapure water was added into the tube with a soil–water ratio of 1:3.5 (*m*/*v*). The tube was shaken at room temperature for 30 min and centrifuged after echocardiography. The pH and salinity in the supernatant were detected using a multiparameter water analysis device (M600L Muti-parameter Analyzer, Shanghai INESA Scientific Instrument C., Ltd., Shanghai, China).

High-throughput microbial screening: According to the standard protocol, sample DNA was extracted using the FastDNA^TM^Spin Kit for Soil (MP Biomedicals, Santa Ana, CA, USA). PCR was performed using the archaeal primers AR109F/AR915R and the bacterial primers BA27F/BA907R, targeting the V4–V5 region of 16S rRNA [40]. The reaction parameters were as follows: pre-denaturation at 94 °C for 2 min; 94 °C denaturation for 30 s; annealing at 55 °C for 30 s. The reaction lasted for 25 cycles at 72 °C for 1 min. PCR products were detected via gel electrophoresis and then cut and purified for high-throughput sequencing. The Library was prepared based on the TruSeq Nano DNA LT Library Prep Kit of Illumina. The End Repair Mix2 kit was used to excise the base-protruding 5′ end of DNA, complete the 3′ end, and add a phosphate group to the 5′ end. The NGS platform Illumina was used for sequencing and the specific sequencing instrument was novaSeq.

The Quantitative Insights Into Microbial Ecology (QIIME) version 1.7.0 pipeline (http://www.Qiime.org, accessed on 1 May 2022) was used to process raw sequencing data with the default parameters [41]. Briefly, the representative sequences from each OTU were defined by a 97% identity threshold level, after which chimeric and low-quality reads were removed. Using the Ribosomal Database Project (RDP) classifier [42], the taxonomic classification of each OTU was assigned. The average relative abundance (%) of the predominant genus-level taxonomy in each sample was assessed by comparing the assigned sequence number of a particular taxon to the total obtained sequence number. To clarify microbial community differences, alpha diversity, PCoA, and Venn were computed with R [40]. The original sequencing data were deposited at the National Center for Biotechnology Information (NCBI) with the accession numbers SRP479735 and SRP479733.

#### Normalized Vegetation Index

The satellite remote sensing data in this study were obtained from the 16 m resolution data of the WFV sensor of China’s Gaofen-1 satellite from the China Centre for Resources Satellite Data and Application (https://data.cresda.cn/#/2dMap, accessed on 5 May 2022) (GF-1 WFV). We downloaded the data from 2019 to 2022 in the study area. The normalized vegetation index (NDVI) was calculated as follows:(1)$$NDVI=\frac{\rho_{NIR}-\rho_{RED}}{\rho_{NIR}+\rho_{RED}}$$

*ρ_RED_* and *ρ_NIR_* represent the images of the red and near-infrared bands, respectively.

## 5. Conclusions

This study showed that salinity levels and the season comprise the core factors affecting the composition of soil flora in the intertidal restoration area. By planting *Suaeda salsa* on saline land, the soil’s salinity and organic carbon environment were effectively improved. However, the restored plants have not yet formed a complete self-sustaining and positive cycle of succession. Continuing investments and technological innovation are still needed in order to restore beach vegetation within the area.

## Figures and Tables

**Figure 1 plants-13-00213-f001:**
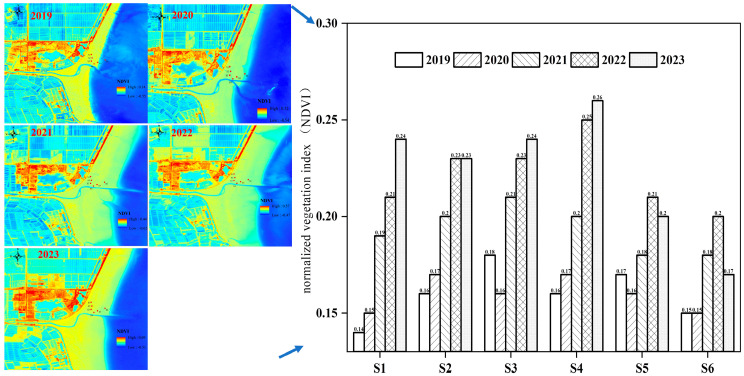
The normalized vegetation index (NDVI) in the study area between 2019 and 2020. S1: Supratidal no-vegetation zone; S2: supratidal vegetative zone; S3: intertidal vegetative zone; S4: intertidal no-vegetation zone in the Yongfeng River Estuary; S5: intertidal vegetative transition zone in the Yongfeng River Estuary; S6: intertidal vegetative zone in the Yongfeng River Estuary.

**Figure 2 plants-13-00213-f002:**
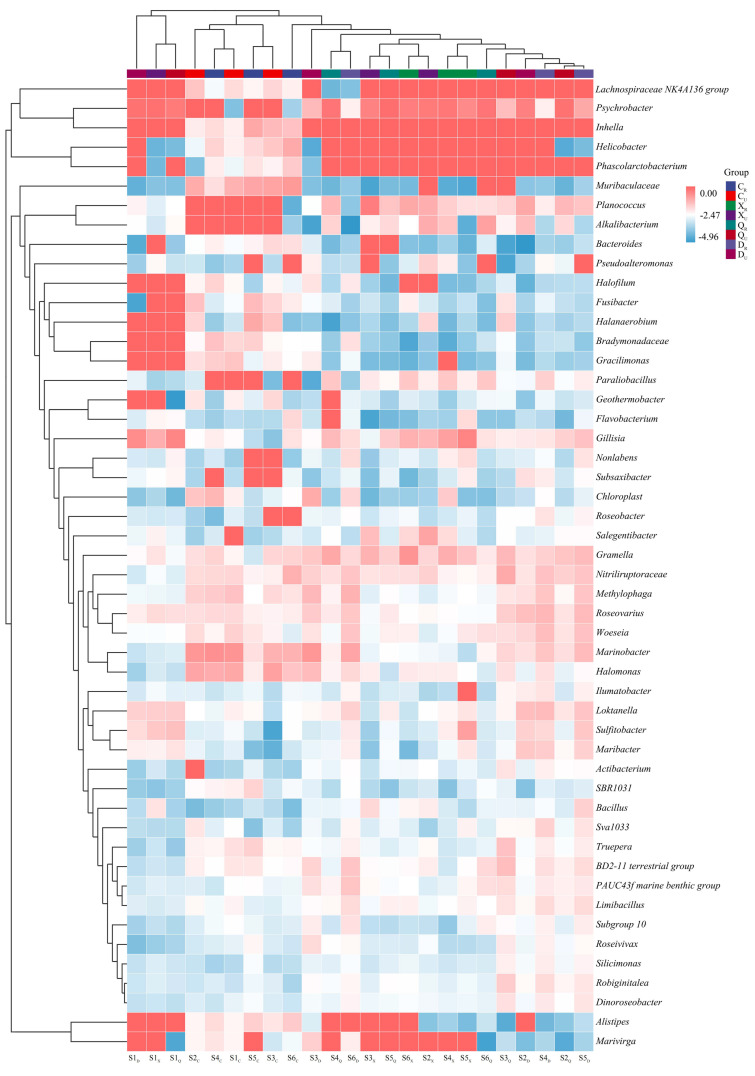
High-throughput bacterial assays in the study area. S1: Supratidal no-vegetation zone; S2: supratidal vegetative zone; S3: intertidal vegetative zone; S4: intertidal no-vegetation zone in the Yongfeng River Estuary; S5: intertidal vegetative transition zone in the Yongfeng River Estuary; S6: intertidal vegetative zone in the Yongfeng River Estuary. C/_C_: Spring; X/_X_: Summer; Q/_Q_: Autumn; D/_D_: Winter. _R_: Vegetative zone; _U_: No-vegetation zone.

**Figure 3 plants-13-00213-f003:**
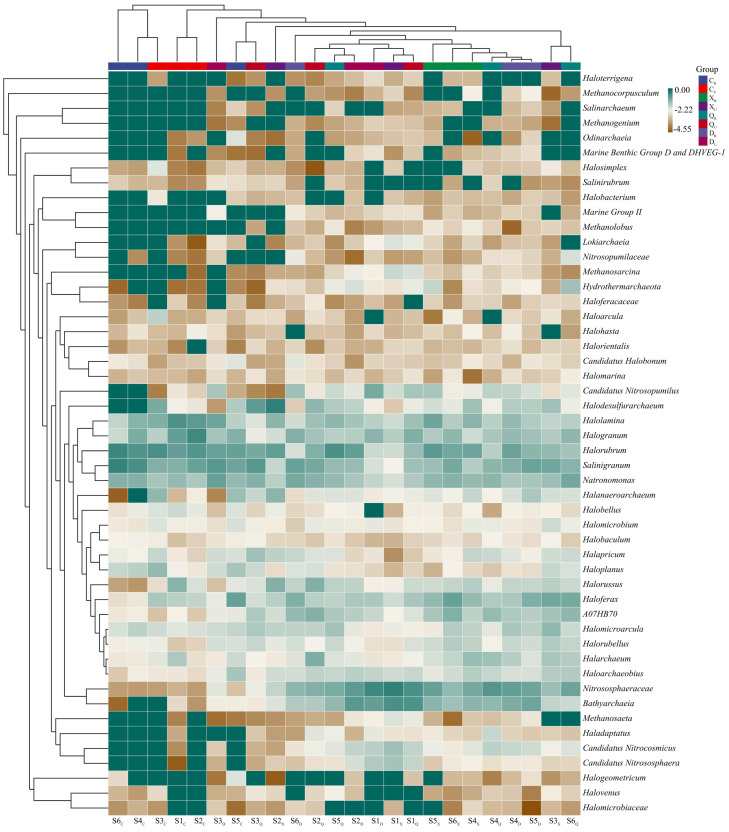
The high-throughput archaea in the study area. S1: Supratidal no-vegetation zone; S2: supratidal vegetative zone; S3: intertidal vegetative zone; S4: intertidal no-vegetation zone in the Yongfeng River Estuary; S5: intertidal vegetative transition zone in the Yongfeng River Estuary; S6: intertidal vegetative zone in the Yongfeng River Estuary. C/_C_: Spring; X/_X_: Summer; Q/_Q_: Autumn; D/_D_: Winter. _R_: Vegetative zone; _U_: No-vegetation zone.

**Figure 4 plants-13-00213-f004:**
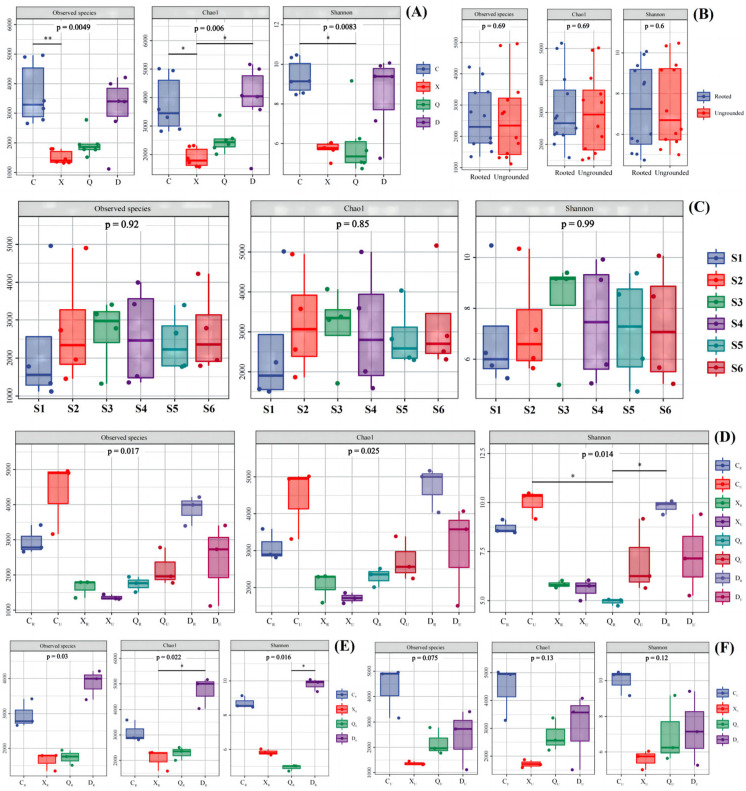
α diversity of bacteria in study area. (**A**) Samples of different seasons; (**B**) Samples of vegetative/no-vegetation zone; (**C**) Samples in different sites during the year; (**D**) Samples of different seasons in vegetative/no-vegetation zone; (**E**) Samples of different seasons in vegetative zone; (**F**) Samples of different seasons in no-vegetation zone. S1: Supratidal no-vegetation zone; S2: supratidal vegetative zone; S3: intertidal vegetative zone; S4: intertidal no-vegetation zone in the Yongfeng River Estuary; S5: intertidal vegetative transition zone in the Yongfeng River Estuary; S6: intertidal vegetative zone in the Yongfeng River Estuary. C: Spring; X: summer; Q: autumn; D: winter. _R_: Vegetative zone; _U_: no-vegetation zone. The * means *p* < 0.05 and ** means *p* < 0.01.

**Figure 5 plants-13-00213-f005:**
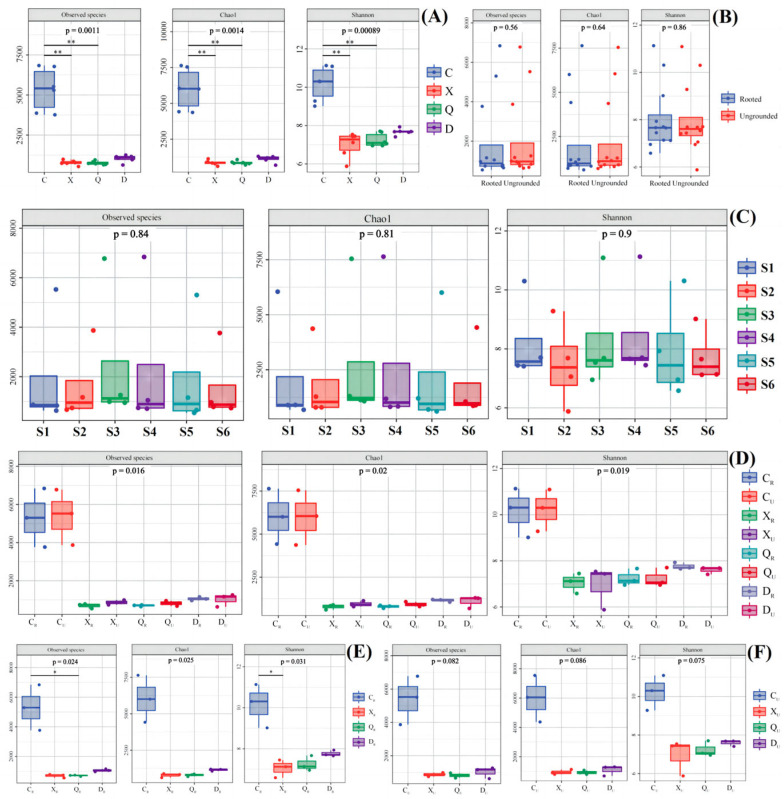
α diversity of archaea in the study area. (**A**) Samples of different seasons; (**B**) Samples of vegetative/no-vegetation zone; (**C**) Samples in different sites during the year; (**D**) Samples of different seasons in vegetative/no-vegetation zone; (**E**) Samples of different seasons in vegetative zone; (**F**) Samples of different seasons in no-vegetation zone. S1: Supratidal no-vegetation zone; S2: supratidal vegetative zone; S3: intertidal vegetative zone; S4: intertidal no-vegetation zone in the Yongfeng River Estuary; S5: intertidal vegetative transition zone in the Yongfeng River Estuary; S6: intertidal vegetative zone in the Yongfeng River Estuary. C: Spring; X: summer; Q: autumn; D: winter. _R_: Vegetative zone; _U_: no-vegetation zone. The * means *p* < 0.05 and ** means *p* < 0.01.

**Figure 6 plants-13-00213-f006:**
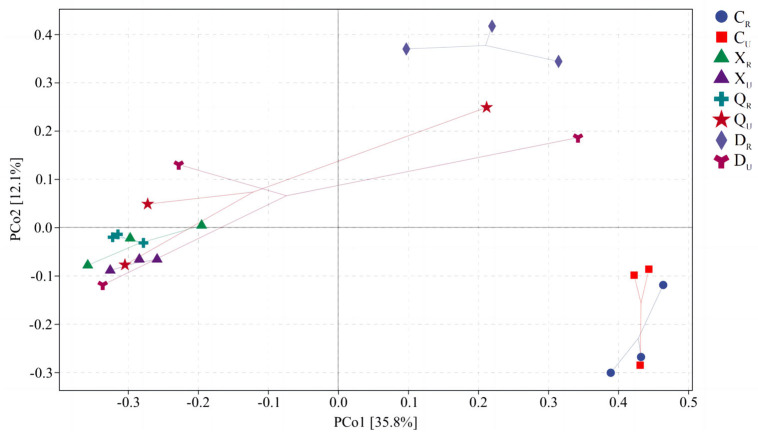
Principal coordinates analysis for bacteria. S1: Supratidal no-vegetation zone; S2: supratidal vegetative zone; S3: intertidal vegetative zone; S4: intertidal no-vegetation zone in the Yongfeng River Estuary; S5: intertidal vegetative transition zone in the Yongfeng River Estuary; S6: intertidal vegetative zone in the Yongfeng River Estuary. C: Spring; X: summer; Q: autumn; D: winter. _R_: Vegetative zone; _U_: no-vegetation zone.

**Figure 7 plants-13-00213-f007:**
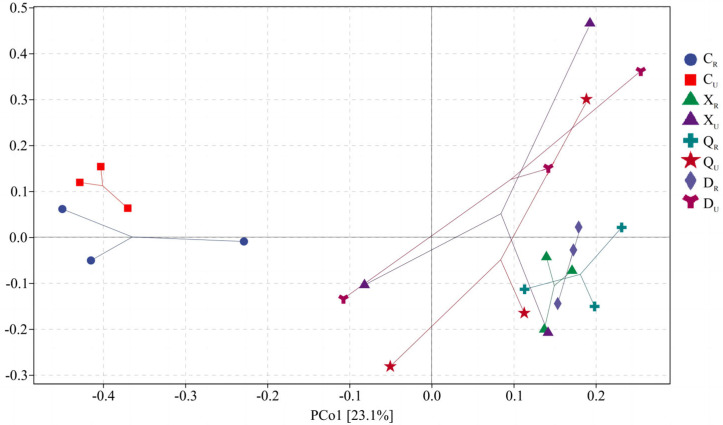
Principal coordinate analysis for archaea. S1: Supratidal no-vegetation zone; S2: supratidal vegetative zone; S3: intertidal vegetative zone; S4: intertidal no-vegetation zone in the Yongfeng River Estuary; S5: intertidal vegetative transition zone in the Yongfeng River Estuary; S6: intertidal vegetative zone in the Yongfeng River Estuary. C: Spring; X: summer; Q: autumn; D: winter. _R_: Vegetative zone; _U_: no-vegetation zone.

**Figure 8 plants-13-00213-f008:**
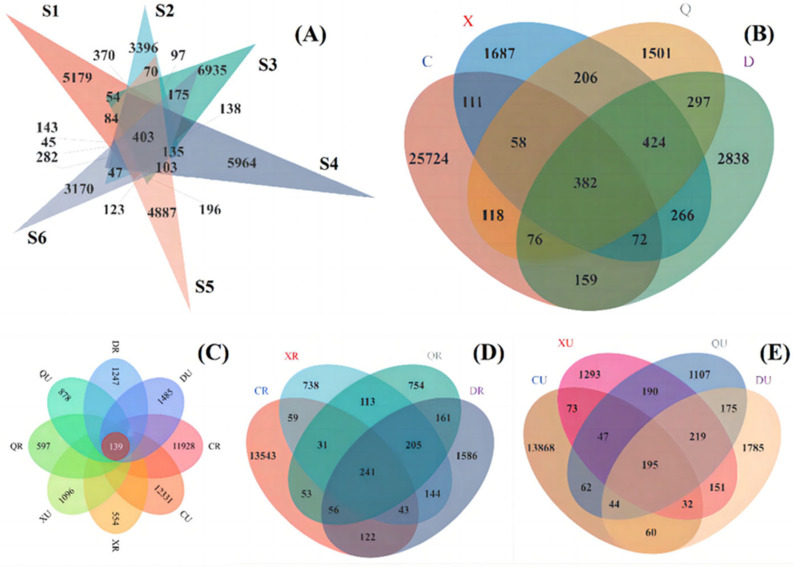
Venn analysis based on seasons and rooting conditions revealed the number of bacteria. (**A**) Samples in different sites during the year; (**B**) Samples of different seasons; (**C**) Samples of different seasons in vegetative/no-vegetation zone; (**D**) Samples of different seasons in vegetative zone; (**E**) Samples of different seasons in no-vegetation zone. S1: Supratidal no-vegetation zone; S2: supratidal vegetative zone; S3: intertidal vegetative zone; S4: intertidal no-vegetation zone in the Yongfeng River Estuary; S5: intertidal vegetative transition zone in the Yongfeng River Estuary; S6: intertidal vegetative zone in the Yongfeng River Estuary. C: Spring; X: summer; Q: autumn; D: winter. R: Vegetative zone; U: no-vegetation zone.

**Figure 9 plants-13-00213-f009:**
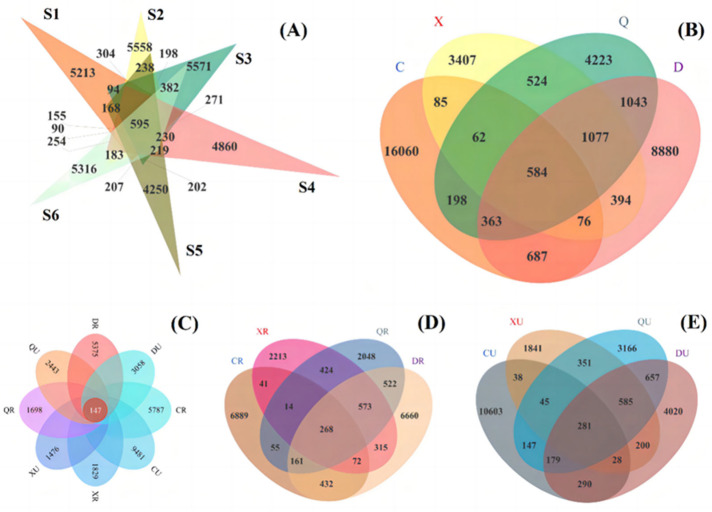
Venn analysis based on seasons and rooting conditions revealed the number of archaea. (**A**) Samples in different sites during the year; (**B**) Samples of different seasons; (**C**) Samples of different seasons in vegetative/no-vegetation zone; (**D**) Samples of different seasons in vegetative zone; (**E**) Samples of different seasons in no-vegetation zone. S1: Supratidal no-vegetation zone; S2: supratidal vegetative zone; S3: intertidal vegetative zone; S4: intertidal no-vegetation zone in the Yongfeng River Estuary; S5: intertidal vegetative transition zone in the Yongfeng River Estuary; S6: intertidal vegetative zone in the Yongfeng River Estuary. C: Spring; X: summer; Q: autumn; D: winter. R: Vegetative zone; U: no-vegetation zone.

**Figure 10 plants-13-00213-f010:**
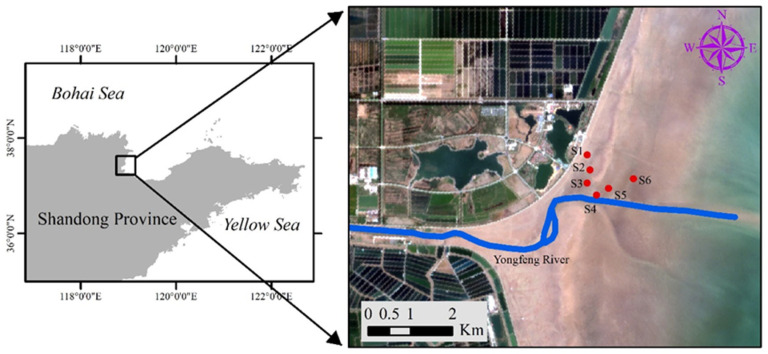
The study area in the Bohai restoration project. The red dots denote the monitor station. S1: Supratidal no-vegetation zone; S2: supratidal vegetative zone; S3: intertidal vegetative zone; S4: intertidal no-vegetation zone in the Yongfeng River Estuary; S5: intertidal vegetative transition zone in the Yongfeng River Estuary; S6: intertidal vegetative zone in the Yongfeng River Estuary.

**Table 1 plants-13-00213-t001:** The displacement distance (in kilometers) of the centroid of aquaculture ponds in each province between 1984 and 2002. S1: Supratidal no-vegetation zone; S2: supratidal vegetative zone; S3: intertidal vegetative zone; S4: intertidal no-vegetation zone in the Yongfeng River Estuary; S5: intertidal vegetative transition zone in the Yongfeng River Estuary; S6: intertidal vegetative zone in the Yongfeng River Estuary.

	Plant Cover	Inundation	pH				Salinity			
			Spring	Summer	Autumn	Winter	Spring	Summer	Autumn	Winter
S1	uncovered	intermittent freshwater	7.92	7.87	7.93	7.96	32.89	32.31	32.80	32.45
S5		no	8.17	8.13	8.18	8.15	45.34	45.21	45.31	45.28
S2		no	8.36	8.25	8.32	8.30	53.63	52.95	53.65	53.59
S3	covered	no	8.45	8.40	8.47	8.42	45.26	45.19	45.31	44.98
S4		no	8.29	8.26	8.31	8.25	30.23	30.15	30.25	30.20
S6		perennial seawater	8.30	8.27	8.35	8.29	30.98	30.82	30.95	30.85

## Data Availability

Data are contained within the article.

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
