# Peer review of "Effect of Intertidal Vegetation (Suaeda salsa) Restoration on Microbial Diversity in the Offshore Areas of the Yellow River Delta"

_plants, 2024, doi:10.3390/plants13020213_

Round 1

Reviewer 1 Report (Previous Reviewer 1)

Comments and Suggestions for Authors

I think I have reviewed this paper entitled “Effects of intertidal Suaeda salsa vegetation eco-restoration on microbial community diversity of coastal zonein November. I have given the opinion that the article was accepted for publication after minor revisions last time. Now, I think this manuscript could be accepted by Plants in the current form. Here are my reasons:

(1) After reading the article in current version carefully, I realized that this work is much upgraded from the previous version, especially the sequencing part of the method. The authors fill in all the microbial information and give a detailed account of the sampling and experimental procedures. The title has also been scientifically modified to become more rational and fit the work and the language has been further enhanced.

(2) This work is very interesting, and the result was an important addition to the theory and practice of reflecting on the relationship between plants and microbiology and human restoration activities in coastal area. It is important to realize that research in this area is relatively scarce.

(3) In this work, the microbial community changes after the restoration of coastal beach wetlands in the whole year were systematically analyzed. It also clarified the actual ability of salinity and plants to influence bacteria and archaea respectively. Those results provided basic data for the screening of strains for the subsequent restoration and modification.

(4) The study combined the results of the geoconfidence study with the structure of the soil microbial community. The situation of restoration and its impacts for the soil was practically and comprehensively valuated. The actual restoration is reflected at the micro level. It could give the directions and gauge for the subsequent vegetation restoration and modification in coastal area.

Therefore, I think this manuscript could be accepted in the current form.

Comments on the Quality of English Language

Minor editing of English language required

Author Response

Dear reviewer,

Thank you very much for taking the time to review this manuscript. Thank you for effort and give the accepted opinion for my manuscript. 

Yours sincerely,

Kai Liu

Reviewer 2 Report (New Reviewer)

Comments and Suggestions for Authors

oofly reading all the documents provided by authors,I found that the manuscript by Wang et al.,which was submitted to the Special Issue of Plants edited by Professor HB SHAO is quite interesting by showing some aspects as concluded below:

1-It for the first time reported detailedly the effect of intertidal vegetation (Suaeda salsa) restoration on microbial diversity in the offshore areas of the Yellow River Delta for the coastal zone in China ,and this will provide potential significance for eco-restorating global coastal ecosystem.

2-The paper also provided complete information forMarinobacterand Halomonas predominantce in the bacterial community during the eco-restoring stage.This will provide good beneficial bacteria selection for regulating plant-microbe interaction and improving rhizosphere in salt-affected soils;

3-The paper also studied the bacterial community relationship by the advanced sequencing and provided important guide for managing  coastal eco-envronmental  basing on vegetation NDVI,which is the first report under coastal intertidal vegetation (Suaeda salsa) type.

Now,the title is more scientific after modification.The methodology section is reliable and can be replicated,whcih has been presented and organized well in good English. The paper content is suitable for Plants journal and has covered more parts of the submitted topic.I suggested that the current version of the paper be accepted for publishing.

Author Response

Dear reviewer,

Thank you very much for taking the time to review this manuscript. Thank you for effort and give the accepted opinion for my manuscript. 

Yours sincerely,

Kai Liu

Reviewer 3 Report (Previous Reviewer 4)

Comments and Suggestions for Authors

The text of the presented research has been edited according to the proposals contained in the previous review. It is also worth considering the presentation of the purpose of the research. It is worth highlighting the research objectives, e.g. by listing the issues.

Line 582, section 4.2 Sample: please give information about number of samples not only the total weight of collected soil

Author Response

For research article

Response to Reviewer 3 Suggestions

1. Summary

Thank you very much for taking the time to review this manuscript. Thank you very much for giving the opinion that the paper has been accepted by minor revisions. And thank you for your reminder on the manuscript. We have improved the manuscript exactly as your reminder. Please find the detailed responses below and the corrections highlighted in the re-submitted files.

2. Questions for General Evaluation

Reviewer’s Evaluation

Response and Revisions

Does the introduction provide sufficient background and include all relevant references?

Yes/Can be improved/Must be improved/Not applicable

Are all the cited references relevant to the research?

Yes/Can be improved/Must be improved/Not applicable

Is the research design appropriate?

Yes/Can be improved/Must be improved/Not applicable

Are the methods adequately described?

Yes/Can be improved/Must be improved/Not applicable

Are the results clearly presented?

Yes/Can be improved/Must be improved/Not applicable

Are the conclusions supported by the results?

Yes/Can be improved/Must be improved/Not applicable

3. Point-by-point response to Comments and Suggestions for Authors

1.     Comments 1: Line 582, section 4.2 Sample: please give information about number of samples not only the total weight of collected soil.

Response 1: Thank you very much for your very helpful reminder on my manuscript. We have not clearly expressed the number of samples here. We have highlighted the number of samples " Three parallel samples were collected simultaneously " in the Lines 556-557 and mark in red.

Yours sincerely,

Kai Liu

This manuscript is a resubmission of an earlier submission. The following is a list of the peer review reports and author responses from that submission.

Round 1

Reviewer 1 Report

Comments and Suggestions for Authors

The paper “The effect of intertidal vegetation (Suaeda salsa) restoration on microbial diversity” conducted a complete study on the soil microbial diversity of the ecological restoration of the coastal wetland in the Yellow River Delta, and the content was rich. Finally, it found the vulnerability of the initial growth and self-maintenance of the salinity, as well as the reality that the impact of microbial diversity in the bottom tidal beach is less than the salinity. However, the following problems still exist in the research paper and need to be corrected by the author.

1. The topic suggests adding the Yellow River Delta because the research scope of this paper does not involve other coastal wetlands.

2. This paper uses the means of ground credit science, but it is not reflected in the subsequent content, and it is suggested to modify it.

3. The content of the article and the pictures of bacteria and archaea format are not unified, and it is recommended to unify the Latin format.

4. There are problems in the format and language, so it is suggested to modify it, including the reference format, carefully.

5. As the vegetation involved in the article is maintained but limited to one year's data, increasing the amount of data is recommended.

Comments on the Quality of English Language

Minor editing of English language required.

Author Response

Response to Reviewer 1 Comments

1. Summary

2. Questions for General Evaluation

Reviewer’s Evaluation

Response and Revisions

Does the introduction provide sufficient background and include all relevant references?

Yes/Can be improved/Must be improved/Not applicable

Are all the cited references relevant to the research?

Yes/Can be improved/Must be improved/Not applicable

Is the research design appropriate?

Yes/Can be improved/Must be improved/Not applicable

Are the methods adequately described?

Yes/Can be improved/Must be improved/Not applicable

Are the results clearly presented?

Yes/Can be improved/Must be improved/Not applicable

Are the conclusions supported by the results?

Yes/Can be improved/Must be improved/Not applicable

3. Point-by-point response to Comments and Suggestions for Authors

Comments 1: All the presentation should be checked for the text including tables, paying attention to punctuation mark.

Response 1: Thank you for pointing this out. We agree with this comment. Therefore, we have checked and modified the punctuation in page 3, paragraph 1 and line 105; page 3, paragraph 1 and line 107; page 5, paragraph 1 and line 140; page 7, paragraph 1 and line 182; page 8, paragraph 3 and line 215; page9, paragraph 2 and line238; page 10, paragraph 2 and line 251; page 10, paragraph 3 and line 253; page 11, paragraph 1 and line 270; page 11, paragraph 4 and line 285; page 12, paragraph 2 and line 305; page 15, paragraph 6 and line 469; page 16, paragraph 2 and line 483; page 16, paragraph 3 and line 485; page 16, paragraph 5 and line 497.

Comments 2: All the cited references should be checked out for the format, pages number and abbreviations.

Response 2: Thank you for pointing this out. We agree with this comment. Therefore, we have checked and modified the cited references form page 17-19, line 533 to 606.

Comments 3: Some key references in Nature Comm, Plants, Microorganisms and others for the past 3 years should be cited for strengthening the significance. Updating the references.

Response 3: Thank you for pointing this out. We agree with this comment. Therefore, we have added some key references in Nature Comm, Plants, Microorganism for the past 3 years in page 19, line 539 to 544; page 19, line 608 to 620.

Comments 4: Language revision is also advisable.

Response 4: Thank you for pointing this out. We agree with this comment. Therefore, we have checked and modified the paper in page 1, paragraph 6, line 30, 32 and 35; page 3, paragraph1, line 106-107; page 5, paragraph 2, line 143-145; page 5, paragraph 2, line 143-145; page 7, paragraph 2, line 193-195; page 8, paragraph 1, line 202-203; page 8, paragraph 2, line 209-211; page 8, paragraph 2, line 214-215; page 10, paragraph 1, line 247-251; page 11, paragraph 3, line 277-278; page 11, paragraph 3, line 281-282; page 11, paragraph 3, line 281-282; page 13, paragraph 2, line 335-338.

Reviewer 2 Report

Comments and Suggestions for Authors

Suaeda salsa vegetation is salt-tolerant and adapted to the salt-affected environment, especially coastal zone. This paper is interesting in terms of studying the effects of primary restoration on microbial communities in the Yellow River Delta by taking Suaeda salsa vegetation as the target,which is a good story and report for the recent years. The submission has been written in good order and logic,in which authors systemically studied the vegetation –soil interactions by determining microbial activities, which influence soil properties and further ecological functions for the first time. Overall, the paper can be accepted after minor revisions as I commented  on below:

1 All the presentation should be checked for the text including tables, paying attention to punctuation mark;

2 All the cited references should be checked out for the format, pages number and abbreviations;

3 Some key references in Nature Comm, Plants, Microorganisms and others for the past 3 years should be cited for strengthening the significance. Updating the references.

Comments on the Quality of English Language

Language revision is also advisable.

Author Response

Response to Reviewer 2 Comments

1. Summary

2. Questions for General Evaluation

Reviewer’s Evaluation

Response and Revisions

Does the introduction provide sufficient background and include all relevant references?

Yes/Can be improved/Must be improved/Not applicable

Are all the cited references relevant to the research?

Yes/Can be improved/Must be improved/Not applicable

Is the research design appropriate?

Yes/Can be improved/Must be improved/Not applicable

Are the methods adequately described?

Yes/Can be improved/Must be improved/Not applicable

Are the results clearly presented?

Yes/Can be improved/Must be improved/Not applicable

Are the conclusions supported by the results?

Yes/Can be improved/Must be improved/Not applicable

3. Point-by-point response to Comments and Suggestions for Authors

Comments 1: The topic suggests adding the Yellow River Delta because the research scope of this paper does not involve other coastal wetlands.

Response 1: Thank you for pointing this out. We agree with this comment. Therefore, we have added the Yellow River Delta in the topic in Page 1 and Line 2.

Comments 2: This paper uses the means of ground credit science, but it is not reflected in the subsequent content, and it is suggested to modify it.

Response 2: Thank you for your comments. We have added the lines for the ground credit science. And this research is based the result of geoinformatics, so the following content is the detail information about soil microbes community as reflecting in the satellite.

Comments 3: The content of the article and the pictures of bacteria and archaea format are not unified, and it is recommended to unify the Latin format.

Response 3: Thank you for pointing this out. We agree with this comment. Therefore, we have unify the Latin format in article in the past 3 years in page 1, paragraph 4, line 11-13, 15-16; page 1, paragraph 4, line 11-13, 15-16;

Comments 4:  There are problems in the format and language, so it is suggested to modify it, including the reference format, carefully.

Response 4: Thank you for your advise. We agree with this comment. Therefore, we have checked and modified the format in page 1, paragraph 4, line 11- 16; page 4, Paragraph 1 line 133-139; page 4, paragraph 2 line 145, 149-153; page 6, paragraph 2, line 159-172, 179; page 6, paragraph 2, line 159-172, 179; page 13, paragraph 4, line 343, 346, 355-360, 365; page 13, paragraph 5, line 367, 371-373, 377-378; page 14, paragraph 1, line 379-383, 385-386; page 14, paragraph 2, line393-394, 407-408, 411, and 415; page 14, paragraph 1, line 379-383, 385-386;

Comments 5: As the vegetation involved in the article is maintained but limited to one year's data, increasing the amount of data is recommended.

Response 5: Thank you for your advise, we have planed the following research proposal based on the data in this research to give more information about microbial diversity in vegetation (Suaeda salsa) restoration areas.

Comments 6:  Minor editing of English language required.

Response 6: Thank you for your advise, we have edited accordingly in page 1, paragraph 6, line 30, 32 and 35; page 3, paragraph1, line 106-107; page 5, paragraph 2, line 143-145; page 5, paragraph 2, line 143-145; page 7, paragraph 2, line 193-195; page 8, paragraph 1, line 202-203; page 8, paragraph 2, line 209-211; page 8, paragraph 2, line 214-215; page 10, paragraph 1, line 247-251; page 11, paragraph 3, line 277-278; page 11, paragraph 3, line 281-282; page 11, paragraph 3, line 281-282; page 13, paragraph 2, line 335-338.

Reviewer 3 Report

Comments and Suggestions for Authors

The problem with this manuscript is that the authors are using words that simply do not exist in the English language. I have no idea what the authors mean by alkali pontic, alkali poncho, or alkali ponts, but these words do not exist in the English language. A poncho is a blanket with a slit in the middle so that it can be slipped over the head and worn as a sleeveless garment. I am sure that is not what the authors mean, and I have no idea what they are talking about.

Comments on the Quality of English Language

Impossible to understand

Author Response

Response to Reviewer 3 Comments

1. Summary

2. Questions for General Evaluation

Reviewer’s Evaluation

Response and Revisions

Does the introduction provide sufficient background and include all relevant references?

Yes/Can be improved/Must be improved/Not applicable

Are all the cited references relevant to the research?

Yes/Can be improved/Must be improved/Not applicable

Is the research design appropriate?

Yes/Can be improved/Must be improved/Not applicable

Are the methods adequately described?

Yes/Can be improved/Must be improved/Not applicable

Are the results clearly presented?

Yes/Can be improved/Must be improved/Not applicable

Are the conclusions supported by the results?

Yes/Can be improved/Must be improved/Not applicable

3. Point-by-point response to Comments and Suggestions for Authors

Comments: The problem with this manuscript is that the authors are using words that simply do not exist in the English language. I have no idea what the authors mean by alkali pontic, alkali poncho, or alkali ponts, but these words do not exist in the English language. A poncho is a blanket with a slit in the middle so that it can be slipped over the head and worn as a sleeveless garment. I am sure that is not what the authors mean, and I have no idea what they are talking about.

Response: Thank you very much for your helpful comments, we have modified the presentation with highlight, such as the name of Suaeda salsa, reference and some sentences in line 17, 53, 56-57, 76, 97, 104, 122, 126, 127, 322, 325, 330-333, 430, 450, 520.

Reviewer 4 Report

Comments and Suggestions for Authors

Lines 44, 487: Suaeda salsa should be written with italics

Line 109: The Figure 1 is not very clear. Maybe it would be better to compile Figure 1 and 10

Please in the title of the figures explain the abbreviations S1…..S6. The same in the case of Table 1

The research paid particular attention to the biodiversity of microbial communities. However, apart from the table with data on the properties of the studied soils, there are no detailed analyzes of their impact on soil properties. Therefore, it is difficult to draw conclusions about the impact of planting Suaeda salsa on improving soil properties and microbial communities. If there are previous studies, it is worth mentioning them here.

Overall, the research is interesting and discusses important aspects of the functioning of intertidal ecosystems, with particular emphasis on microbial communities and, in particular, the importance of biodiversity within these communities.

Author Response

Response to Reviewer 4 Comments

1. Summary

2. Questions for General Evaluation

Reviewer’s Evaluation

Response and Revisions

Does the introduction provide sufficient background and include all relevant references?

Yes/Can be improved/Must be improved/Not applicable

Are all the cited references relevant to the research?

Yes/Can be improved/Must be improved/Not applicable

Is the research design appropriate?

Yes/Can be improved/Must be improved/Not applicable

Are the methods adequately described?

Yes/Can be improved/Must be improved/Not applicable

Are the results clearly presented?

Yes/Can be improved/Must be improved/Not applicable

Are the conclusions supported by the results?

Yes/Can be improved/Must be improved/Not applicable

3. Point-by-point response to Comments and Suggestions for Authors

Comments 1: Lines 44, 487: Suaeda salsa should be written with italics.

Response 1: Thank you for pointing this out. This comment is very important to improve the quality of our articles. We agree with this comment. Therefore, we checked all the manuscript and modified the italics in page 1, paragraph 5, line 24; page 1, paragraph 7, line 44; page 4, paragraph 1, line 134; page 16, paragraph 1, line 476; page 17, paragraph 3, line 518; page 19, paragraph 2, line 589.

Comments 2: Line 109: The Figure 1 is not very clear. Maybe it would be better to compile Figure 1 and 10.

Response 2: Thank you for pointing this out. This comment is very important to improve the quality of our articles. We agree with this comment. Therefore, we uploaded high-quality version of Figure 1 in the supplementary material.

Comments 3: Please in the title of the figures explain the abbreviations S1…..S6. The same in the case of Table 1.

Response 3: Thank you for pointing this out. We agree with this comment. Therefore, we have added the abbreviations S1-S6 in title of the Figure 1, lines 112-115; the Table 1, lines 137-139; the Figure 2, lines 151-155; the Figure 3, lines 196-200; the Figure 4, lines 234-237; the Figure 5, lines 259-262; the Figure 10, lines 513-516.

Comments 4:  The research paid particular attention to the biodiversity of microbial communities. However, apart from the table with data on the properties of the studied soils, there are no detailed analyzes of their impact on soil properties. Therefore, it is difficult to draw conclusions about the impact of planting Suaeda salsa on improving soil properties and microbial communities. If there are previous studies, it is worth mentioning them here.

Overall, the research is interesting and discusses important aspects of the functioning of intertidal ecosystems, with particular emphasis on microbial communities and, in particular, the importance of biodiversity within these communities.

Response 4: Thank you for your advice. We agree with this comment. Therefore, we have checked and modified some discussion in page 16, paragraph 1, line 471-475 and added reference 39 in 663-664.

Round 2

Reviewer 1 Report

Comments and Suggestions for Authors

I agree to accept the publication of the manuscript.

Author Response

Thank you very much for your very helpful comments on my manuscript. Thank you again for reviewing my manuscript.

Reviewer 2 Report

Comments and Suggestions for Authors

The author revised the MS according to the reviwers' advice, and i am satisfied with the present version.

Author Response

(The authors gave the same response as above.)

Reviewer 3 Report

Comments and Suggestions for Authors

I have reviewed manuscript plants-2696825 by Wang and Liu. There is a serious problem in the Methods section of the manuscript. The description of the DNA analysis consists of three sentences as follows: “The sample DNA was extracted using FastDNATM SPIN Kit. All the operation steps follow the standard instruction flow provided by the reagent company. The extracted DNA samples are put into dry ice, and then sent to the testing company for sequencing no more than 72 h.”  I think it is generally understood that the methods section of a manuscript should provide sufficient detail that someone else could repeat the procedure, and this description does not come remotely close to fulfilling that expectation. For example, there is no mention of what part of the DNA was analyzed, and there is no mention of the primers that were used. There is no mention of what rationale was used to identify organisms to the genus level (and in one case to the family level). This paper is about impacts on microbial diversity, but it would be impossible for another person to repeat the assessment of microbial diversity without far more information about the methods used to assess microbial diversity.

Author Response

Response to Reviewer 3 Comments

1. Summary

2. Questions for General Evaluation

Reviewer’s Evaluation

Response and Revisions

Does the introduction provide sufficient background and include all relevant references?

Yes/Can be improved/Must be improved/Not applicable

Are all the cited references relevant to the research?

Yes/Can be improved/Must be improved/Not applicable

Is the research design appropriate?

Yes/Can be improved/Must be improved/Not applicable

Are the methods adequately described?

Yes/Can be improved/Must be improved/Not applicable

Are the results clearly presented?

Yes/Can be improved/Must be improved/Not applicable

Are the conclusions supported by the results?

Yes/Can be improved/Must be improved/Not applicable

3. Point-by-point response to Comments and Suggestions for Authors

Comments 1: I have reviewed manuscript plants-2696825 by Wang and Liu. There is a serious problem in the Methods section of the manuscript. The description of the DNA analysis consists of three sentences as follows: “The sample DNA was extracted using FastDNATM SPIN Kit. All the operation steps follow the standard instruction flow provided by the reagent company. The extracted DNA samples are put into dry ice, and then sent to the testing company for sequencing no more than 72 h.”  I think it is generally understood that the methods section of a manuscript should provide sufficient detail that someone else could repeat the procedure, and this description does not come remotely close to fulfilling that expectation. For example, there is no mention of what part of the DNA was analyzed, and there is no mention of the primers that were used. There is no mention of what rationale was used to identify organisms to the genus level (and in one case to the family level). This paper is about impacts on microbial diversity, but it would be impossible for another person to repeat the assessment of microbial diversity without far more information about the methods used to assess microbial diversity.

Response 1: Thank you for pointing this out. This comment is very important to improve the quality of our articles. We agree with this comment. Therefore, we have added and modified some the expressions in page 17, paragraph 4, and line536-542; page 18, paragraph 1 and line 543-551; some references were also added in page 20, reference 40-42, line 665-672.